# Ovarian cancer prevention through opportunistic salpingectomy during abdominal surgeries: A cost-effectiveness modeling study

**Angela Kather**[1,2]*, **Habib Arefian**[3¤], **Claus Schneider**[4], **Michael Hartmann**[3],
**Ingo B. Runnebaum**[1,2,5]*

1 Department of Gynecology and Reproductive Medicine, University Hospital Jena, Friedrich Schiller University Jena, Jena, Germany, 2 Zentrum für Alternsforschung Jena—Aging Research Center Jena, Jena, Germany, 3 Hospital Pharmacy, University Hospital Jena, Friedrich Schiller University Jena, Jena, Germany, 4 Department of General, Visceral and Vascular Surgery, Jena University Hospital, Friedrich Schiller University Jena, Jena, Germany, 5 RU21 GmbH, Jena, Germany

¤ Current address: BIG direkt gesund, Berlin, Germany
* angela.kather@med.uni-jena.de (AK); ingo.runnebaum@med.uni-jena.de (IBR)

## Abstract

### Background

There is indication that the fallopian tubes might be involved in ovarian cancer pathogenesis and their removal reduces cancer risk. Hence, bilateral salpingectomy during hysterectomy or sterilization, so called opportunistic salpingectomy (OS), is gaining wide acceptance as a preventive strategy. Recently, it was discussed whether implementation of OS at other gynecologic surgery, e.g., cesarean section, endometriosis excision or myomectomy and even at non-gynecologic abdominal surgery such as cholecystectomy or appendectomy for women with completed family could be feasible. This modeling analysis evaluated the clinical and economic potential of OS at gynecologic and abdominal surgeries.

### Methods and findings

A state transition model representing all relevant health states (healthy, healthy with hysterectomy or tubal ligation, healthy with other gynecologic or non-gynecologic abdominal surgery, healthy with hysterectomy and salpingectomy, healthy with salpingectomy, healthy with hysterectomy and salpingo-oophorectomy, ovarian cancer and death) was developed and informed with transition probabilities based on inpatient case numbers in Germany (2019). Outcomes for women aged 20–85 years were simulated over annual cycles with 1,200,000 million individuals. We compared four strategies: (I) OS at any suitable abdominal surgery, (II) OS only at any suitable gynecologic surgery, (III) OS only at hysterectomy or sterilization, and (IV) no implementation of OS. Primary outcome measures were prevented ovarian cancer cases and deaths as well as the incremental cost-effectiveness ratio (ICER). Volume of eligible interventions in strategy I was 3.5 times greater than in strategy III (286,736 versus 82,319). With strategy IV as reference, ovarian cancer cases were

**Data availability statement:** All relevant data are within the manuscript and its Supporting information file.

**Funding:** The author(s) received no specific funding for this work.

**Competing interests:** I have read the journal''s policy and the authors of this manuscript have the following competing interests: IBR is a member of the Ovarian Tumor Committee of the German Gynecologic Oncology Group (AGO) and member of the writing committee responsible for releasing the 'S3 Guideline on Diagnosis, Treatment, and Follow-up of Malignant Ovarian Tumors.' The other authors declare no conflicts of interest.

**Abbreviations:** BRCA, BReast CAncer gene; GDP/C, gross domestic product per capita; HR, hazard ratio; ICER, incremental cost-effectiveness ratio; LY, life year; OS, opportunistic salpingectomy; PARP, Poly (Adenosine diphosphate-Ribose) Polymerase; QALY, quality-adjusted life year; STICs, serous tubal intraepithelial carcinomas.

reduced by 15.34% in strategy I, 9.78% in II, and 5.48% in III. Setting costs for OS to €216.19 (calculated from average OS duration and operating room minute costs), implementation of OS would lead to healthcare cost savings as indicated by an ICER of €−8,685.50 per quality-adjusted life year (QALY) gained for strategy I, €−8,270.55/QALY for II, and €−4,511.86/QALY for III. Sensitivity analyses demonstrated stable results over a wide range of input parameters with strategy I being the superior approach in the majority of simulations. However, the extent of cancer risk reduction after OS appeared as the critical factor for effectiveness. Preventable ovarian cancer cases dropped to 4.07% (I versus IV), 1.90% (II versus IV), and 0.37% (III versus IV) if risk reduction would be <27% (hazard ratio [HR] > 0.73). ICER of strategies I and II was lower than the 2× gross domestic product per capita (GDP/C) (€94,366/QALY, Germany 2022) within the range of all tested parameters, but strategy III exceeded this threshold in case-risk reduction was <35% (HR > 0.65). The study is limited to data from the inpatient sector and direct medical costs.

## Conclusions

Based on our model, interdisciplinary implementation of OS in any suitable abdominal surgeries could contribute to prevention of ovarian cancer and reduction of healthcare costs. The broader implementation approach demonstrated substantially better clinical and economic effectiveness and higher robustness with parameter variation. Based on a lifetime cost saving of €20.89 per capita if OS was performed at any suitable abdominal surgery, the estimated total healthcare cost savings in Germany could be more than €10 million annually.

### Author summary

#### Why was this study done?

- Ovarian cancer is a rare but deadly disease, because many cases are diagnosed in advanced stages. This is due to a lack of efficient early diagnosis and prevention strategies.

- Treatment of ovarian cancer is burdensome for patients and costly for the healthcare system.

- Researchers have found that the most common and aggressive types of ovarian cancer often originate from the fallopian tubes.

- During past 15 years, there was a trend to recommend removal of both fallopian tubes at the opportunity of gynecologic pelvic surgery, such as hysterectomy or tubal sterilization. Ovaries are left in place for continuing hormone production. This strategy is called "opportunistic salpingectomy (OS)".

- Enabling fallopian tube removal also at other abdominal surgery such as cholecystectomy or appendectomy in women who have completed their family is discussed among clinicians.

- So far, it is not clear whether performing OS can noticeably reduce ovarian cancer cases, provide lifetime health benefits, or impact overall healthcare costs in European countries such as Germany.

## What did the researchers do and find?

- We developed a mathematical model that describes the chance of having a surgery with opportunity for fallopian tube removal for each women and the chance of reducing her risk for ovarian cancer.

- By applying this model to the entire female population of Germany, we showed that 5% of ovarian cancer cases could be prevented if fallopian tubes were removed during every hysterectomy and tubal sterilization. This prevention rate could be increased to 15% if they were removed during every suitable abdominal surgery in women who have completed their families, leading to significant health improvements and extended healthy years of life.

- Broad implementation of opportunistic fallopian tube removal could lead to healthcare cost savings.

## What do these findings mean?

- This study shows that removing both fallopian tubes during suitable abdominal surgeries can be an effective way to prevent ovarian cancer at the population level.

- The findings of this study could help health policymakers and insurance providers to calculate appropriate compensation for the costs of fallopian tube removal. This, in turn, could support the acceptance of the procedure within the population, particularly in societies like Germany, where tubal sterilization and other contraceptive methods are generally not covered by insurance. In contrast, in the United States of America, tubal sterilization is often compensated by various plans and is more commonly performed.

- The analysis was limited to direct medical costs. Socioeconomic costs and benefits were not considered or calculated, suggesting that broader impacts, including potential benefits, were not assessed. Furthermore, data for outpatient procedures are not available in Germany, limiting the analysis to the number of inpatient surgical interventions.

## Introduction

Worldwide, ovarian cancer is the third most common gynecologic cancer representing 22% of cancer cases and shows a high mortality rate of 66% [1]. Average age of onset for ovarian cancer is 69 years [2]. Due to demographic changes, absolute numbers of ovarian cancer cases and deaths are likely to rise considerably in high income countries over the next few decades. Unfortunately, screening approaches have been unsuccessful thus far [3]. Consequently, prevention strategies are of high importance.

At the beginning of the new millennium, scientific evidence was accumulating for an extra-ovarian origin of many histologic subtypes of ovarian cancer. Cancer precursor cells probably originate from or are transported through fallopian tubes [4–10]. For serous tubal intraepithelial carcinomas (STICs), it was shown that they carry the same tumor-specific genetic alterations as the concomitant ovarian cancer lesions [10]. This would imply that timely removal of the fallopian tubes (salpingectomy) could potentially confer protection. Indeed, population-based cohort studies have found a significantly reduced risk after salpingectomy in women of the general population [11–13]. Hysterectomy [11,14] or tubal ligation [11,15] also resulted in reduced risk; however, to a lesser extent.

Based on these compelling observations, the concept of opportunistic salpingectomy (OS) at the time of hysterectomy and in lieu of tubal ligation for permanent contraception was developed and adopted by gynecologic surgeons in some countries, such as United States of America (USA), Canada, Australia, Japan, Taiwan, Great Britain, Denmark, Austria and Germany [16–20]. However, randomized-controlled trials are still missing to prove the effect of OS and are unlikely to be conducted due to the low prevalence of ovarian cancer and the long-time period between the typical age of intervention and the average age of disease onset. Nevertheless, first data from British Columbia indicate that OS can be effective and practicable as an ovarian cancer prevention strategy [21] and safety of OS was shown in several studies [22–26].

Recent findings demonstrate that OS is feasible also at non-gynecologic abdominal surgery such as cholecystectomy [27], considerably expanding the range of occasions for OS. However, OS is currently not always performed at the time of gynecologic surgery and almost never at the time of other elective surgeries. Furthermore, in several countries such as Germany, no concept for financial coverage of the additional time and cost for OS has been developed for statutory health insurance systems.

Several research groups have utilized decision-analytic models to investigate the clinical and cost effectiveness of OS [28–31]. However, only few studies have considered the whole life span of women, precisely deciphering the combination of the chance for having an abdominal surgery with opportunity for OS and risk of ovarian cancer [32,33]. To our knowledge, up to now there is no population-level lifetime modeling study addressing all opportunities for OS for a European country and only one for the USA [33], which; however, does not provide an elaborate comparison of all possible implementation strategies.

This study aims to evaluate the potential of OS for ovarian cancer prevention at population level and its impact on healthcare costs in a lifetime model including all eligible abdominal surgeries based on real-world data. Different OS implementations strategies are compared to a scenario without OS, considering prevented ovarian cancer cases and deaths as primary clinical outcome measures and the incremental cost-effectiveness ratio (ICER) as the economic outcome measure.

## Methods

### Basic considerations and assumptions

It is not currently recommended that women with an average risk of developing ovarian cancer undergo surgery solely as a means of preventing the disease through salpingectomy. OS is thought to be performed exclusively on occasion of a gynecologic or non-gynecologic abdominal intervention that is medically necessary, or if a women decides to have permanent contraception. Consequently, the number and type of surgeries with opportunity for OS is the same in all strategies analyzed in this study. Therefore, costs and risks of these surgeries were not considered.

Ovarian cancer risk reduction after removal of fallopian tubes might be different for histologic subtypes [12,21], due to the diverse origin of precursor lesions [34]. Furthermore, there are women with increased ovarian cancer risk, e.g., BReast CAncer gene (BRCA) mutation carriers. However, all data used to inform the model do not distinguish between histological subtypes or between women with varying cancer risk. Consequently, the results of the model represent estimates for the impact of OS on overall ovarian cancer incidence, mortality and healthcare costs at the population level. Women with increased cancer risk require specialized consultation and are recommended to undergo bilateral salpingo-oophorectomy between the ages of 35 and 45 years [35].

## Decision-analytic model and compared strategies

A state-transition model was developed according to the respective guidelines [36–38], representing all possible health states necessary for this analysis, such as healthy, healthy with non-gynecologic surgery, healthy with hysterectomy, healthy with tubal ligation, healthy with salpingectomy, healthy with hysterectomy and salpingectomy, healthy with hysterectomy and bilateral salpingo-oophorectomy, healthy with other gynecologic surgery, ovarian cancer and death (absorbing state) (Fig 1).

Four strategies were compared (Fig 1). In strategy I (gynecologic + non-gynecologic), OS is conducted at any eligible gynecologic surgery (hysterectomy and sterilization at any age; cesarean section, ovarian cyst removal, inpatient endometriosis surgery, open abdominal or laparoscopic myomectomy and uterus fixation starting from age 40 years) and non-gynecologic abdominal surgery (cholecystectomy, hernia closure, bariatric surgery, planned uncomplicated appendectomy) starting from 40 years of age (Table A in S1 Text). Total case number of these surgeries performed in 2019 in Germany was n = 286,736 [39]. In strategy II (gynecologic), OS is conducted at any gynecologic surgery (n = 132,255), while in strategy III OS is conducted only

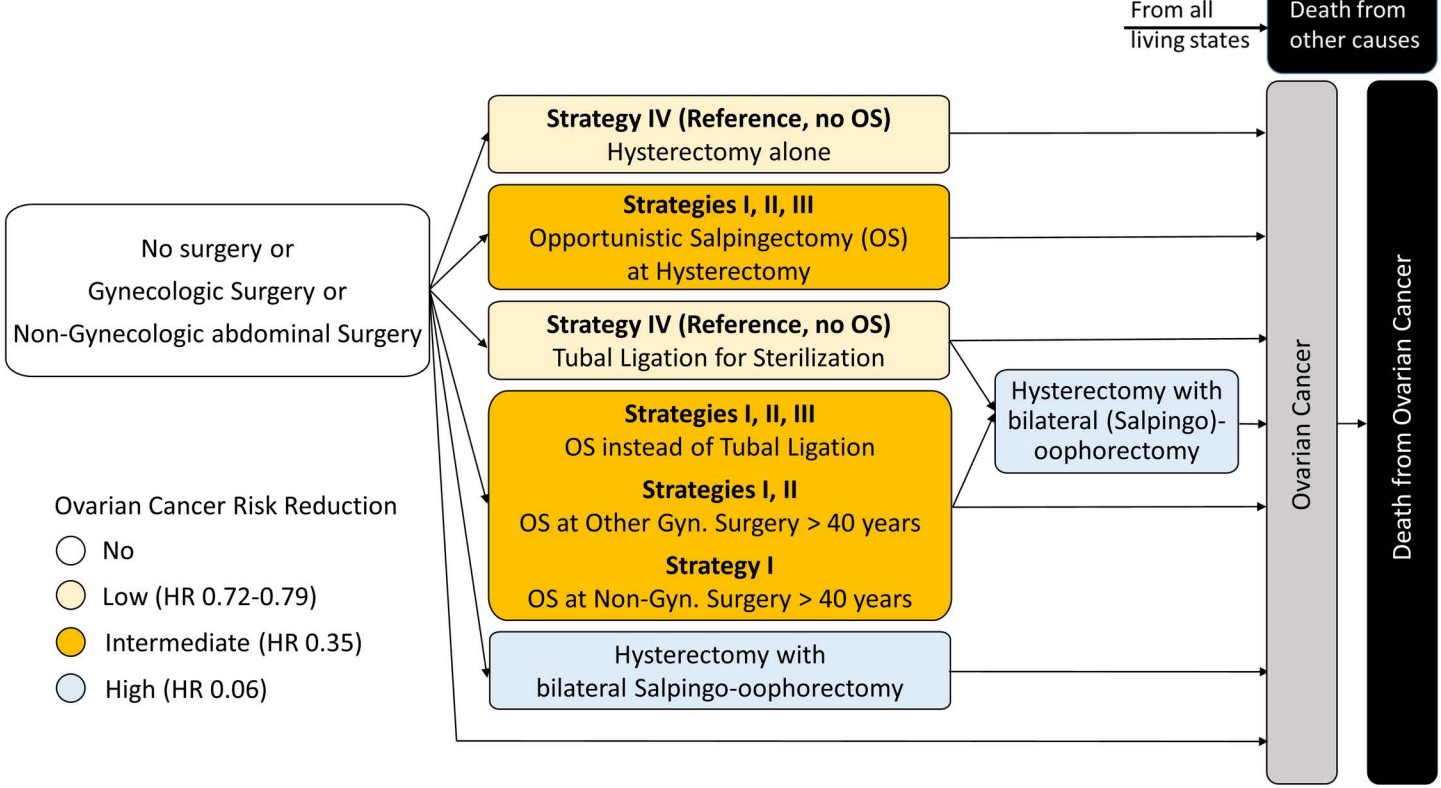

**Fig 1. The state-transition diagram of the opportunistic salpingectomy (OS) decision-analytic model, presented here in simplified form, represents the possible health states associated with surgery or the absence of it.** OS is thought to be performed exclusively on occasion of a gynecologic or non-gynecologic abdominal intervention that is medically necessary, or if a women decides to have permanent contraception. Women stay in each state until a relevant event occurs. Transition is only allowed if it leads to increased ovarian cancer risk reduction. Four strategies are compared: (I) OS performed at any suitable abdominal surgical intervention (Gyn. + Non-Gyn.), (II) OS performed only at any suitable gynecologic surgery (Gyn.), (III) OS performed only at hysterectomy or as an alternative to tubal ligation for sterilization (current practice in some countries) and (IV) no implementation of OS (reference strategy). Women can finally end up in the states "death of other causes" or "death of ovarian cancer" as absorbing states. Hazard ratios (HRs) for ovarian cancer risk after the respective interventions according to Falconer and colleagues [11]. Gyn., gynecologic; Non-Gyn., non-gynecologic.

at hysterectomy and sterilization ($n$ = 82,319), which represents current practice in at least 13 countries, including the USA, Canada, Great Britain, Germany, France, Austria, Turkey, Australia, Taiwan and Japan [17,18,20,40]. Strategy IV is the reference strategy with no OS performed.

All women start at age 20 years in the healthy state and transition through various health states in annual cycles (Fig 1). A woman will stay in the healthy state until she undergoes a surgery with an opportunity for OS. Depending on the chosen strategy, OS is either performed or not. If OS is performed, she will transition to one of the appropriate states after salpingectomy (healthy with salpingectomy or healthy with hysterectomy and salpingectomy). In the healthy with salpingectomy state, the woman can still undergo hysterectomy with bilateral oophorectomy, resulting in further reduction of ovarian cancer risk. If OS is not performed, the woman will go to the appropriate state after surgery without OS (healthy with non-gynecologic surgery, healthy with hysterectomy, healthy with tubal ligation or healthy with other gynecologic surgery). From these states, transitions are only allowed if they result in decreased cancer risk. In all states, women can die because of background mortality or get ovarian cancer based on the risk associated with the respective state. After diagnosis, woman can either survive or die according to age-dependent ovarian cancer mortality. We assumed that 10 years after diagnosis mortality returns to background.

## Model parameters

**Transition probabilities.** Age-adjusted annual transition probabilities (Fig 2 and Tables B–I in S1 Text) were calculated according to the equation $P = I/N_0$ [41] based on pre-pandemic case numbers (year 2019) obtained from the Federal Statistical Office of Germany [39] and the German Center for Cancer Registry Data [42]. Ovarian cancer-related mortality was subtracted from background mortality to avoid double counting.

Tubal sterilizations are predominantly performed in an ambulatory setting as a same day procedure [43] and are not covered by the health insurance in Germany. Case numbers for outpatient private surgeries are currently not recorded in Germany. Hence, transition probabilities for sterilization were calculated based on inpatient case numbers, which represent the age distribution, and calibrated in order to obtain the observed proportion of women in Germany, who claim to have undergone sterilization. We decided to conservatively choose a rate of 5%, which is the proportion observed over all age groups [44]. This was achieved with 2.5-times increased inpatient values.

**Effect measures and utilities.** Risk reduction after hysterectomy (0.79), tubal occlusion (0.72), bilateral salpingectomy (0.35) and hysterectomy with bilateral salpingo-oophorectomy (0.06) was based on hazard ratios (HRs) from the literature [11]. A latency period was implemented, assuming that risk reduction after OS is effective starting from 5 years after surgery [11,13].

Presumably, any surgery with opportunity for OS will not result in compromised quality of life for more than a few weeks and age-adjusted population average utilities [45] were addressed to all healthy states. Utility for each year in the state "Ovarian cancer" (utility = 0.61) was calculated as a weighed value between non-advanced (23%, utility = 0.81) and advanced (77%, utility = 0.55) disease [2] according to literature [32,46,47]. It was assumed that implementation of OS does not alter the complication rate of the respective surgery [48].

**Costs and cost-effectiveness threshold.** For cost-effectiveness analysis, the perspective of the German healthcare system was adopted, and only direct medical costs were considered. Costs of the interventions, on which occasion OS could be performed, were not included, because their indication and incidence is not different between the investigated strategies.

Extra surgical time for OS is on average 13 min [27] resulting in €216.19 costs based on average operating room minute price in Germany [49]. Costs of ovarian cancer treatment

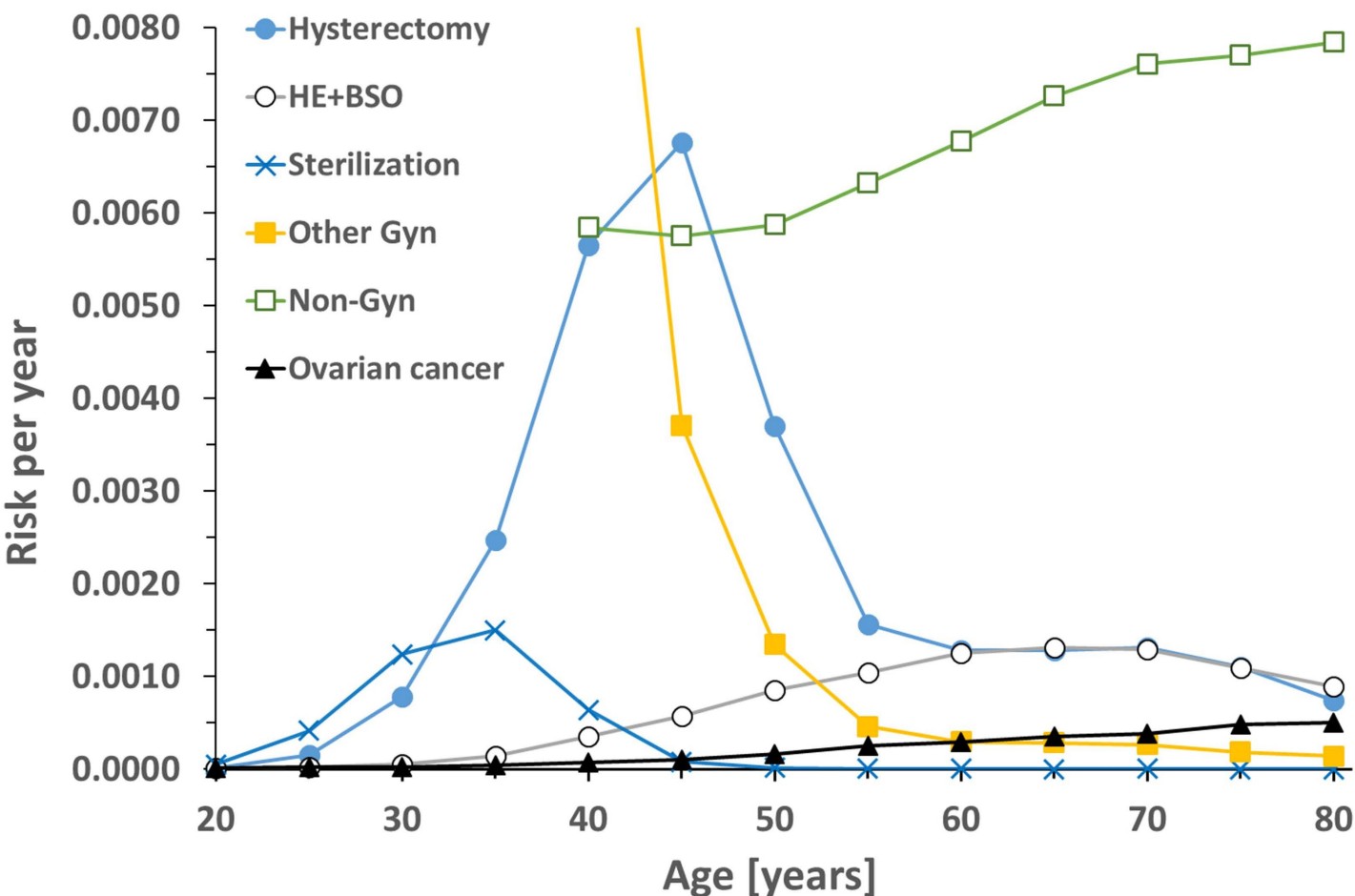

**Fig 2. Age-dependent yearly probabilities (risk) for surgery with occasion for opportunistic salpingectomy and risk of ovarian cancer.** Calculated from pre-pandemic case numbers (year 2019) obtained from the Federal Statistical Office of Germany and the German Center for Cancer Registry Data. Non-gynecologic (non-gyn) abdominal surgery included cholecystectomy, hernia closure, bariatric surgery and scheduled uncomplicated appendectomy starting from 40 years of age. Other gynecologic (other gyn) surgery includes cesarean section, ovarian cyst removal, inpatient endometriosis surgery, open abdominal or laparoscopic myomectomy and uterus fixation starting from age 40 years. HE+BSO, hysterectomy with bilateral salpingo-oophorectomy.

were derived from literature [50] and adjusted for the proportion of early and advanced stages, resulting in one-time €30,133.22 for primary treatment. Follow-up costs of annually €36,366.49 over 5 years include surveillance as well as diagnosis/treatment of recurrence and recently approved maintenance therapy with Poly (Adenosine diphosphate-Ribose) Polymerase (PARP) inhibitors and Bevacizumab calculated from literature values [50] and pharmacy retail price 2023 based on guideline-compliant treatment regimen [35] (Tables J and K in S1 Text). In the event of dying from ovarian cancer, a final amount for palliative care of €12,103.00 incurs once [50].

Because no official threshold is defined for Germany, cost effectiveness of the compared strategies was evaluated in relation to the GDP/C in Germany (year 2022), which was €47,183 [51].

## Analysis

The model was built and analyzed in TreeAge Pro Healthcare Version 2024 (TreeAge Software, LLC, Williamstown, Massachusetts, USA). Because some transitions were dependent on

history (survival and follow-up costs after ovarian cancer diagnosis, entry into force of effect of OS), Monte Carlo Microsimulation with half-cycle correction was used for analysis. Concordant with similar models [32,50], cycle length was set to 1 year and analysis was run for 65 cycles for a lifetime analysis covering ages 20–85 years.

**Base case analysis.** Clinical outcome parameters were prevented ovarian carcinoma cases and deaths (%), undiscounted life years (LYs) and quality-adjusted life years (QALYs) gained. A 3% annual discount rate was applied to costs and effects as suggested [38] for calculation of the ICER (incremental costs per incremental QALY gained). The model was run multiple times with increasing number of individuals until stable results were obtained with a simulation cohort of 1,200,000 individuals (Figs A–C in S1 Text).

**Sensitivity analysis.** Transition probabilities were calculated from real-world population level data, so they do not contain the problems associated with parameter estimation (parameter uncertainty) and heterogeneity between individuals [52]. However, from a scenario analysis view, it could have an impact if incidence of surgeries with opportunity for OS would change. For example, the number of hysterectomies almost halved during the past 15 years due to a trend to organ preserving therapy [53]. This is in part, but not fully, compensated by the implementation of Myomectomy into our model. To explore this in more detail, we performed an analysis with 50% incidence of hysterectomy. Furthermore, it is likely that recent improvements in ovarian cancer treatment, e.g., introduction of maintenance therapy, lead to a better prognosis [54,55]. To reflect this, analysis was also performed with a 30% reduced risk for death after ovarian cancer diagnosis. Transition probabilities for sterilization were analyzed for inpatient case numbers as the lower bound in order to explore the effect of the applied calibration. For the upper bound, 7.5-times increased inpatient numbers were used to give the proportion of women (13%) who stated to have undergone sterilization at the ages of 40–49 years [44].

Risk reduction after OS was tested within the range of the 95% confidence interval (0.17–0.73) of the HR [11]. Utility of the ovarian cancer health state was varied between 0.5 and 0.82 according to published standard deviations [46]. Variation of the discount rate was analyzed between 0% and 5% as suggested [38]. Range of possible costs for OS (lower bound €66.52, upper bound €748.35) was calculated based on the published variation in additional operating room time (4–45 min) [27]. Costs for primary therapy of ovarian cancer were varied between 50% and 200% because no information could be found in literature. The lower bound of follow-up costs were set to the value without the recently established maintenance therapy (€2,151.41) [50]. The upper bound was set to 200% of the base case value.

## Results

### Model validation

Regarding ovarian cancer, the model yielded a lifetime risk of 1:86, mortality of 77%, and average age at diagnosis of 64 years, reproducing published epidemiological data [2]. Values of the model for average age were 60 years at non-gynecologic surgery, 46 years at other gynecologic surgery, 33 years at sterilization and 52 years at hysterectomy, closely resembling input values (Fig 2). The prevalence of 14% for hysterectomy in women at age 70 years or older in our model is considerably lower compared to 39.4% found in a survey conducted in Germany during years 2008–2011 [56]. However, all types of hysterectomies were counted in the publication by Prütz and colleagues, while we only included hysterectomies for benign indications. Furthermore, hysterectomy case numbers almost halved since the beginning of the millennium [53], resulting in lower prevalence in the long term.

## Clinical effectiveness

Compared to strategy IV (no OS), ovarian cancer cases could be reduced by 5.48% if strategy III (OS at hysterectomy and sterilization) was applied, 9.78% in strategy II (OS at any gynecologic surgery) and 15.34% in strategy I (OS at any eligible abdominal surgery) (Table 1). With regard to ovarian cancer case numbers in Germany (n = 7,180 in 2020) [57], this would correspond to 395 (strategy III), 703 (strategy II) and 1,099 (strategy I) prevented cases per year. Lifetime risk could be lowered to 1:102 in strategy I. Ovarian cancer related deaths could be decreased by 5.65% (III), 10.27% (II) and 15.99% (I).

The number of salpingectomies which have to be performed to prevent one case of ovarian cancer was lowest in strategy II (n = 209) and highest in strategy III (n = 256), while in strategy I it was n = 232.

**Table 1. Clinical and cost effectiveness of opportunistic salpingectomy in base case simulation with 1,200,000 women over 65 annual cycles (covering ages 20–85 years).**

| Simulation cohort size = 1,200,000 | | I (Gyn + Non-Gyn) | II (Gyn) | III (HE + Steri) | IV (No OS) |
|---|---|---|---|---|---|
| Ovarian cancer | Cases in simulation cohort [n] | 11,762 | 12,534 | 13,131 | 13,893 |
| | Cases prevented [n (%)] | 2,131 (15.34) | 1,359 (9.78) | 762 (5.48) | Ref. |
| | Preventable annual cases in Germany [n] | 1,099 | 703 | 395 | Ref (n = 7,180[a]) |
| | Lifetime risk | 1:102 | 1:96 | 1:91 | 1:86 |
| | Deaths in simulation cohort [n] | 9,007 | 9,620 | 10,115 | 10,721 |
| | Deaths prevented [n (%)] | 1,714 (15.99) | 1,101 (10.27) | 606 (5.65) | Ref. |
| | Preventable annual deaths in Germany [n] | 842 | 541 | 297 | Ref (n = 5,265[a]) |
| | Mean age at diagnosis [years] (SD) | 63.49 (6.40) | 64.02 (6.66) | 64.24 (6.83) | 64.48 (7.05) |
| Salpingectomies | Number of Salpingectomies per prevented ovarian cancer case | 232 | 209 | 256 | – |
| | Average age at OS [years] (SD) | 50.57 (26.47) | 44.15 (19.33) | 43.41 (16.75) | – |
| Life years per capita | Mean (SD), undiscounted | 60.36802 (0.00797) | 60.36389 (0.00797) | 60.35975 (0.00797) | 60.35433 (0.00797) |
| | Gained, undiscounted | 0.01369 | 0.00956 | 0.00542 | Ref. |
| | Mean (SD), discounted[b] | 28.31946 (0.00211) | 28.31867 (0.00211) | 28.31784 (0.00211) | 28.31674 (0.00211) |
| | Gained, discounted[b] | 0.00271 | 0.00192 | 0.00109 | Ref. |
| QALYs per capita | Mean (SD), undiscounted | 51.70536 (0.00647) | 51.70181 (0.00647) | 51.69824 (0.00647) | 51.69358 (0.00647) |
| | Gained, undiscounted | 0.01178 | 0.00822 | 0.00466 | Ref. |
| | Mean (SD), discounted[b] | 24.86762 (0.00176) | 24.86693 (0.00176) | 24.86619 (0.00176) | 24.86522 (0.00176) |
| | Gained, discounted[b] | 0.00240 | 0.00170 | 0.000974 | Ref. |
| Mean lifetime costs per capita[b] | Salpingectomy [€] (SD) | 39.00 (0.05) | 26.78 (0.05) | 18.65 (0.04) | 0 |
| | Treatment of ovarian cancer [€] (SD) | 471.13 (5.31) | 490.11 (5.37) | 507.99 (5.43) | 531.02 (5.52) |
| | Total [€] (SD) | 510.13 (5.31) | 516.90 (5.37) | 526.64 (5.43) | 531.02 (5.52) |
| | Savings [€] | 20.89 | 14.12 | 4.38 | Ref. |
| | Savings [%] | 3.93 | 2.66 | 0.82 | Ref. |
| Cost effectiveness[b] | Cost-effectiveness ratio [€/QALY] | 20.51 | 20.79 | 21.18 | 21.36 |
| | ICER [€/QALY] | −8,685.50 | −8,270.55 | −4,511.86 | Ref. |

Four strategies were compared: (I) OS performed at any suitable abdominal surgical intervention (Gyn + Non-Gyn), (II) OS only at any suitable gynecologic surgery (Gyn), (III) OS only at hysterectomy and in lieu of tubal ligation for sterilization (HE + Steri, current practice in some countries) and (IV) no implementation of OS (reference strategy). Additional costs for salpingectomy were set to €216.19 based on 13 min extraoperating room time [27,49]. Risk reduction after salpingectomy was assumed to be 65% [11] taking effect from 5 years after surgery [13].

[a]Annual ovarian cancer case number 2020 according to German cancer registry [57].

[b]Discount rate 3%.

OS, opportunistic salpingectomy; n, number; SD, standard deviation; Ref., reference; QALY, quality-adjusted life year; ICER, incremental cost-effectiveness ratio.

Without discounting, 0.00542 (III), 0.00956 (II) and 0.01369 (I) LYs and 0.00466 (III), 0.00822 (II) and 0.01178 (I) QALYs could be gained per person compared to strategy IV.

## Cost effectiveness

In base case analysis with an annual discount rate of 3%, OS would result in healthcare cost savings. Strategy I was most cost-effective, followed by strategy II. All OS-strategies were superior to the traditional practice without OS. Cost savings per QALY gained (ICER) would be €−8,685.50/QALY (I versus IV), €−8,270.55/QALY (II versus IV) and €−4,511.86/QALY (III versus IV) (Table 1). Similar results were obtained for cost savings per life year gained (€−7,698.77/LY (I versus IV), €−7,344.83/LY (II versus IV) and €−4,006.21/LY (III versus IV).

## Sensitivity analyses

In one-way deterministic sensitivity analyses, the model showed robust results for variations in transition probabilities for sterilization, hysterectomy and ovarian cancer mortality, utility of ovarian cancer and costs associated with ovarian cancer primary as well as palliative care treatment (Tables L and M in S1 Text).

Reduction of ovarian cancer cases and deaths was observed for all tested parameter variations with stable hierarchy of strategies, demonstrating robust clinical effectiveness of OS (Table L in S1 Text). However, if cancer risk reduction after salpingectomy would be lower than 27% (HR > 0.73), the proportion of ovarian cancer cases that could be prevented by OS would drop to 4.07% for strategy I compared to IV, 1.90% (II compared to IV) and 0.37% (III compared to IV).

With strategy IV as reference, strategy I emerged as the most cost-effective strategy in majority of simulations, except when costs for OS exceeded €394.78 (Fig 3C), when the latency period would be longer than 7.5 years (Fig 3B) and when no discounting would be applied (Table M in S1 Text). Strategy II would be superior in these scenarios; however, with a minimal difference. Strategy III showed inferior cost-effectiveness compared to strategies I and II over the range of variation in all parameters (Fig 3 and Table M in S1 Text).

In many tested scenarios, implementation of OS resulted in healthcare cost savings (negative ICER). However, additional expenditures (positive ICER) would occur if the cost of OS exceeded €332.01 (strategy I), €330.23 (II), and €267.04 (III) (Fig 3C), corresponding to more than 20 min, 19.9 min, and 16.1 min of additional operating time, respectively. A positive ICER was also observed in case that yearly follow-up costs were lower than €18,807 (I), €19,182 (II) and €26,904 (III) (Fig 3D) or if an annual discount rate higher than 5.0% (I), 4.7% (II) and 3.7% (III) was applied (Table M in S1 Text). Furthermore, expenses for the healthcare system would arise if cancer risk reduction by OS was lower than 48% (HR > 0.52, I), 50% (HR > 0.50, II) and 58% (HR > 0.42, III) (Fig 3A) or if the latency period after surgery was longer than 12.7 (I), 13.0 (II) and 8.3 (III) years (Fig 3B).

However, ICER of strategies I and II was lower than the 2 × GDP/C (€94,366/QALY) within the range of all tested parameters and even lower than 1 × GDP/C except in case if risk reduction after salpingectomy would be less pronounced (<28%, HR > 0.72 for I; < 31%, HR > 0.69 for II, Fig 3). For strategy III, the ICER exceeded 2 × GDP/C only in case that risk reduction by OS would be lower than 35% (HR > 0.65, Fig 3A) and if the latency period was longer than 18.4 years (Fig 3B).

Because risk reduction after salpingectomy appeared to be a critical factor for effectiveness of OS, a probabilistic sensitivity analysis was conducted with sampling of risk reduction in combination with a sampling of OS costs (Table N in S1 Text). Compared to strategy IV, strategies I and II showed clinical effectiveness (reduction of ovarian cancer cases) in > 98% of

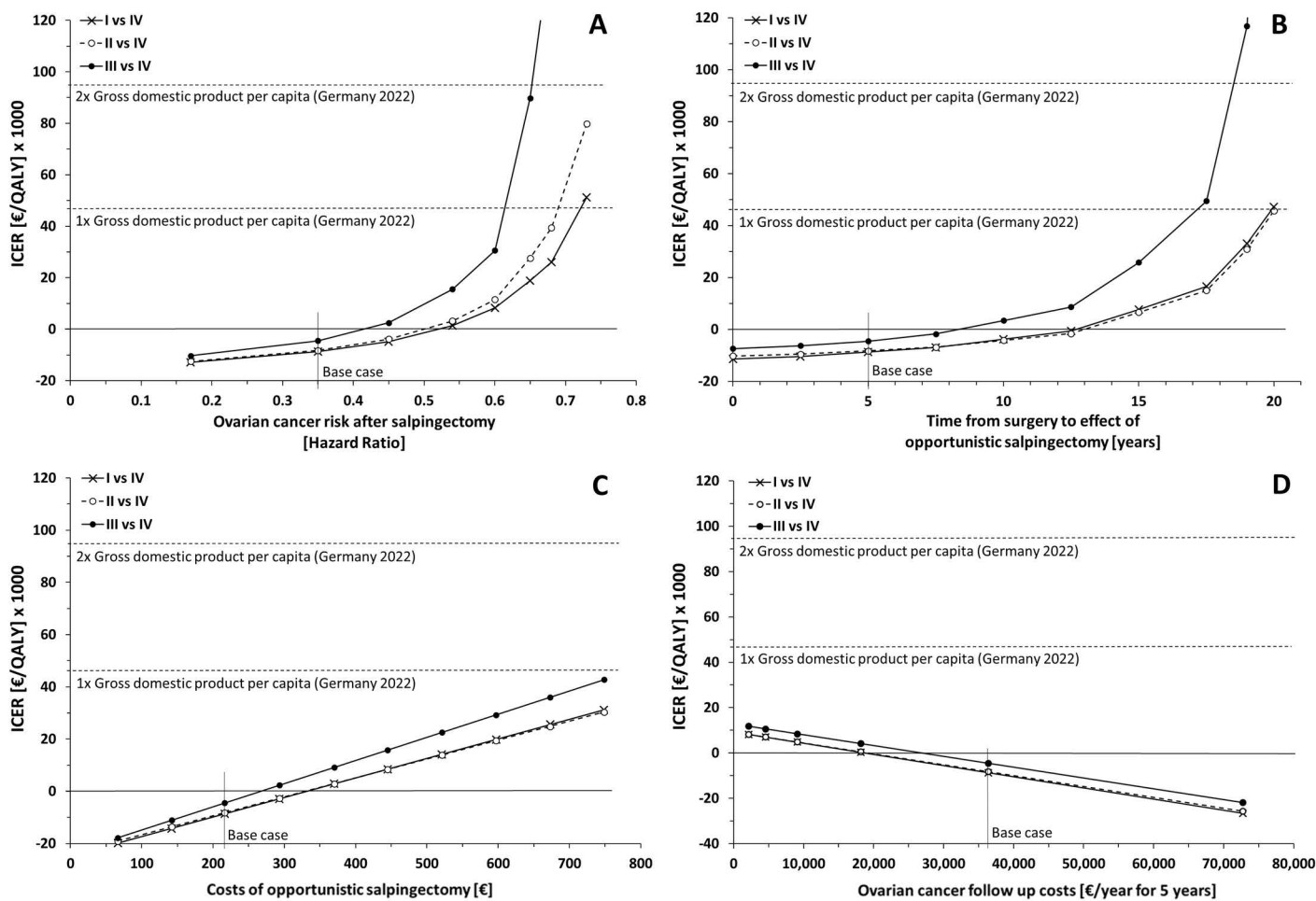

**Fig 3. Results of one-way deterministic sensitivity analyses of the decision-analytic model for opportunistic salpingectomy (OS).** The incremental cost-effectiveness ratio (ICER) was calculated for strategy I (OS at any suitable gynecologic and non-gynecologic abdominal surgery), strategy II (OS at any suitable gynecologic surgery) and strategy III (OS only at hysterectomy and in lieu of tubal ligation for sterilization) compared to strategy IV (no OS) as reference. "Base case" refers to a simulation with base case values for all parameters. Parameters were varied within the range of measures of precision found in the literature. (**A**) Variation of ovarian cancer risk after salpingectomy [11]. (**B**) Variation of time from surgery to effect of OS (latency period) [13]. (**C**) Variation of OS costs [27,49]. (**D**) Variation of ovarian cancer follow-up costs ([50] and own calculations). QALY, quality-adjusted life year. Gross domestic product per capita in Germany was €47,183 in the year 2022 [51].

simulations and an ICER below the 1 × GDP threshold in >98.8% and > 99.4%. Strategy III was clinically effective in > 96.4% and cost-effective in > 95.2% of simulations (Fig D in S1 Text).

## Discussion

The adoption of OS as a clinical practice will necessitate modifying existing recommendations by country-specific expert panels based on calculation of cost-effective reimbursement. Furthermore, reorganization of processes in the operating room with precise cooperation among surgical disciplines and specific training is needed. A precondition for initiating and advancing these changes is availability of reliable information. To our knowledge, this is the first study, which presents a comprehensive, population-based lifetime model for analyzing the clinical and economic effectiveness of OS in preventing ovarian cancer, including a stepwise investigation of eligible gynecologic and non-gynecologic abdominal surgeries.

Feasibility of conducting prospective randomized trials to evaluate effectiveness of OS in preventing ovarian cancer is limited due to the need for long-term observation over decades and very large population samples. These challenges underscore the importance of employing decision analytic methods.

Based on our model, OS could lead to a substantial prevention of ovarian cancer cases and deaths. Implementing OS across all eligible abdominal surgeries results in a pronounced increase in clinical effectiveness compared to the current strategy, which involves OS only at hysterectomy or in lieu of tubal occlusion for sterilization. The interdisciplinary approach demonstrates comparable cost-effectiveness, but offers superior clinical outcomes and greater robustness in sensitivity analyses. In consideration of the fact that ovarian cancer still is a burdensome and deadly disease and no effective early diagnosis strategy is available, prevention of approximately 10% of cases per year would be a great achievement.

Based on €20.89 lifetime cost savings per capita in strategy I, yearly estimated healthcare cost savings could be more than €10 million in Germany given a female population of approximately 42 million and an average life expectancy of 85 years. Our data indicate that reimbursement of OS up to €332 could be possible without additional expenditures for the healthcare system. Partial cost coverage for outpatient sterilization procedures, which currently is a self-payer service, would be desirable and could be justified because of efficient cancer prevention.

Our results are in line with publications by Naumann and colleagues [32,33]. However, the model presented here allows more detailed stepwise analysis of OS implementation strategies. As far as we know, it represents the first framework on this topic based on data from a European country considering costs in Euro. In many European countries, including Germany, the proportion of women choosing sterilization as contraception method after completion of family planning is considerably lower (≤5%) compared to the US (12%) [44,58,59] and because of the trend to organ-preserving therapy, numbers of hysterectomies decline [18]. Therefore, clinical effectiveness of OS at hysterectomy or sterilization is lower in our model compared to Naumann and colleagues [32] (5.7% versus 14.5% prevented deaths), emphasizing the significance of implementation at any eligible abdominal surgery. However, in recent years, there has been an increased demand for tubal sterilization in several countries, possibly due to concerns over a slightly increased risk of breast cancer associated with hormonal contraception.

Some physicians recommend bilateral salpingo-oophorectomy at the time of hysterectomy for patients over the age of 50 years [60], citing its efficacy as the most effective prevention against ovarian cancer [11]. However, the ovaries may still play a residual endocrine role at advanced ages [61]. In contrast, no known function exists for fallopian tubes in the post-reproductive phase, and total bilateral salpingectomy performed during abdominal surgery does not increase the complication rate [27,48]. There might be a risk of a negative impact of salpingectomy on ovarian function if the blood supply to the ovaries is compromised; however, it can be minimized if performed accurately by a surgeon trained in the procedure [62,63]. Several prospective studies are currently underway to properly address this issue [64,65]. A Swedish national register-based randomized non-inferiority trial on salpingectomy, the SALSTER study, has just recently demonstrated that laparoscopic salpingectomy does not increase the risk of complications compared to tubal occlusion up to eight weeks postoperatively, confirming it as a safe alternative [26].

Crucially, OS must only be undertaken once a patient's family planning is complete. Comprehensive consultation about benefits and potential negative consequences of OS before any scheduled abdominal surgery is indispensable to facilitate an informed decision. Use of a structured and validated decision aid [66] might help to standardize counseling and ensure quality. Some studies with a small number of participants indicate that the majority of women

would opt for OS as the permanent contraception method [67–69] after appropriate counseling, because of the presumably lower failure rate and the expected reduction in risk for ovarian cancer.

Regarding the question of an age limit for OS, there is some indication that OS after the age of 50 years might be less effective [13]. However, taking into consideration that ovarian cancer is a very serious disease, OS should not be withheld from women of higher age until further studies have clarified this topic.

Model results were stable over a wide range of input parameters. OS would be cost-effective even if no maintenance therapy for ovarian cancer was applied. Since few years, PARP inhibitors are administered to patients with advanced ovarian cancer for up to 2 years after primary chemotherapy, sometimes in combination with Bevacizumab. This results in profoundly increased costs of more than 100,000 Euro per year of treatment (Table J in S1 Text).

Importantly, the degree of ovarian cancer risk reduction after OS appeared to be a crucial factor for effectiveness. If risk reduction was lower than 35% (HR > 0.65), benefits of OS would be less compelling, calling into question the rationale for OS. Reported HR for bilateral salpingectomy were 0.35, 95% CI 0.17–0.73 [11], and 0.46, 95% CI 0.31–0.67 [13]. However, these data were derived from retrospective population-based studies with small case numbers.

Currently, there are limited data on the feasibility and the readiness of general surgeons to incorporate OS at non-gynecologic abdominal surgeries [70]. Modest adjustments to surgical department processes are sufficient to support interdisciplinary cooperation and provide the essential training for general surgeons. While OS is certainly not technically demanding, the key challenge lies in ensuring surgeons to routinely consider it as an option and dedicate time to adequately consult with patients.

We present a valid and robust model reproducing input data and real epidemiologic numbers. It incorporates comprehensive information based on real-world population data regarding OS at any eligible abdominal surgery for prevention of ovarian cancer. It allows lifetime analysis of effects and costs for four different implementation strategies. However, definitive conclusions should not be drawn solely based on results of this model.

Our analysis is limited to data from the inpatient sector and direct medical costs. Socioeconomic costs were not considered. Likewise, costs of unintended pregnancies, potential earlier menopause due to unintended compromise of the blood supply to the ovaries (which can be avoided with the correct technique) and for pathological examination were not included. However, risk for presence of ovarian cancer precursors (STIC) in fallopian tubes removed during OS in women of the general population is low [71,72], rendering pricy exhaustive systematic pathological examination unnecessary.

The findings of this study contribute information about the possible impact of OS implementation, which could help expert committees, policymakers and insurance providers to refine recommendations, calculate appropriate compensation for the costs of fallopian tube removal and supervise communication and research regarding this topic. This, in turn, could impact on the acceptance of the procedure by doctors and their patients.

OS during eligible abdominal surgeries demonstrated efficacy in reducing the incidence of ovarian cancer, a disease challenging to diagnose in its early stages, and offers healthcare cost savings. Expanding opportunities for OS to a broader range of abdominal interventions beyond already widely accepted hysterectomy and sterilization procedures substantially increased clinical effectiveness and economic robustness. However, effective integration of OS into routine clinical practice of any abdominal surgery necessitates interdisciplinary cooperation and the establishment of suitable financial frameworks. Adequate informed consent processes are indispensable to minimize post-intervention regret and must address potential future fertility concerns as well as the residual risk of earlier menopause.

## Supporting information

**S1 Text.** **Table A:** Surgeries with opportunity for OS included in each strategy. **Table B:** Background mortality for women in Germany 2018–2020. **Table C:** Age-dependent ovarian cancer mortality. **Table D:** Age-dependent risk (transition probability) for diagnosis of ovarian cancer. **Table E:** Age-dependent relative survival rates after ovarian cancer diagnosis. **Table F:** Risk of dying after ovarian cancer diagnosis (hazard = transition probability, *P*). **Tables G and H:** Age-dependent risk (transition probability) for gynecologic surgery with opportunity for OS. **Table I:** Age-dependent risk (transition probability) for non-gynecologic abdominal surgery with opportunity for OS. **Table J:** Recalculation of follow-up (FU) costs including maintenance therapy. **Table K:** Calculation of ovarian cancer primary therapy and follow-up (FU) costs. **Fig A:** Monte Carlo Microsimulation with increasing numbers of individuals until stable low variance was achieved. **Fig B:** Monte Carlo Microsimulation with increasing numbers of individuals until stable ovarian cancer rate was achieved. **Fig C:** Monte Carlo Microsimulation with increasing numbers of individuals until stable ICER was achieved. **Table L:** Results of one-way deterministic sensitivity analysis regarding clinical outcome. **Table M:** Results of deterministic sensitivity analysis regarding health-economic outcome. **Table N:** Characteristics of distributions used for probabilistic sensitivity analysis. **Fig D:** Proportion of simulations giving the indicated percentage of prevented ovarian cancer cases in probabilistic sensitivity analysis.
(DOCX)

## Acknowledgments

We thank R. Wendel Naumann for providing insights into details of his model published in Naumann and colleagues 2021 [https://doi.org/10.1016/j.ajog.2021.03.032].

## Author contributions

**Conceptualization:** Angela Kather, Claus Schneider, Michael Hartmann, Ingo B. Runnebaum.

**Data curation:** Angela Kather.

**Formal analysis:** Angela Kather.

**Investigation:** Angela Kather.

**Methodology:** Habib Arefian, Michael Hartmann.

**Project administration:** Ingo B. Runnebaum.

**Resources:** Ingo B. Runnebaum.

**Supervision:** Michael Hartmann, Ingo B. Runnebaum.

**Validation:** Angela Kather, Habib Arefian, Claus Schneider, Michael Hartmann, Ingo B. Runnebaum.

**Visualization:** Angela Kather.

**Writing – original draft:** Angela Kather, Ingo B. Runnebaum.

**Writing – review & editing:** Habib Arefian, Claus Schneider, Michael Hartmann.

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
