## [Editor Report · Decision Letter 0]

25 Jun 2024

Dear Dr Kather,

Thank you for submitting your manuscript entitled "Turning the Tide on Ovarian Cancer: Enhancing Prevention with Cost-Effective Opportunistic Salpingectomy—A Decision Analytic Assessment Across Abdominal Surgeries" for consideration by PLOS Medicine.

Your manuscript has now been evaluated by the PLOS Medicine editorial staff and I am writing to let you know that we would like to send your submission out for external peer review.

Please re-submit your manuscript within two working days, i.e. by Jun 27 2024.

Feel free to email me at atosun@plos.org or us at plosmedicine@plos.org if you have any queries relating to your submission.

Kind regards,

Alexandra Tosun, PhD

Associate Editor

PLOS Medicine

atosun@plos.org

---

## [Decision Letter · Decision Letter 1]

9 Sep 2024

Dear Dr Kather,

Many thanks for submitting your manuscript "Turning the Tide on Ovarian Cancer: Enhancing Prevention with Cost-Effective Opportunistic Salpingectomy—A Decision Analytic Assessment Across Abdominal Surgeries" (PMEDICINE-D-24-02003R1) to PLOS Medicine. The paper has been reviewed by subject experts and a statistician; their comments are included below and can also be accessed here: [LINK]

As you will see, the reviewers found the manuscript to be interesting, but they raised a number of methodological concerns. After discussing the paper with the editorial team and an academic editor with relevant expertise, I'm pleased to invite you to revise the paper in response to the reviewers' comments. We plan to send the revised paper to some or all of the original reviewers, and we cannot provide any guarantees at this stage regarding publication.

We ask that you submit your revision by Sep 30 2024. However, if this deadline is not feasible, please contact me by email, and we can discuss a suitable alternative.

Don't hesitate to contact me directly with any questions (atosun@plos.org).

Best regards,

Alexandra

Alexandra Tosun, PhD

Associate Editor

PLOS Medicine

atosun@plos.org

Comments from the academic editor:

The comments of reviewers 1 and 2 are critical: in particular, histotype specificity for interventions (only high-grade serous carcinoma is thought to arise from the fallopian tube, other histotypes arise from different tissues and the risk reduction value is not known), number inflation.

The editorial team would like to note that, in line with the comments of reviewer 1, we would prefer that you reduce the number of abbreviations used throughout the manuscript. We also wondered if you could be more explicit about the absolute risk of ovarian cancer in the relevant group to contextualize "reduced by 15.3%" and the like.

Comments from the reviewers:

Reviewer #1: Thank you for allowing me to review this interesting manuscript. My comments are below.

Abstract

Line 28 - In general, the language here should be softened. We assume that the fallopian tubes are involved in ovarian cancer pathogenesis and removal reduces cancer risk based on observational data, but we do not know this for certain. This needs to be rephrased.

Line 44 - Again we do not know that we can prevent ovarian cancer by removing tubes, only reducing risk. please rephrase.

Introduction

Line 61 - Here hysterectomy and tubal ligation are abbreviated, and then these abbreviations are essentially never used again in the manuscript. Either use these abbreviations consistently, or do not define them at all.

Methods

Line 104 - during strategy 4, are the reference surgeries still performed, but with salpingectomy? or are no surgeries performed at all. this needs to be stated.

Line 118 - is the model outlined in a figure? if not this would be helpful to understand how this was set up for the reader as this block of text is challenging to read without a visual. I think this is what is being conveyed in Figure 1 and this should be reference here if so.

Line 140 - why not use the raw numbers obtained from the inpatient data? why inflate it to match the rate observed in Germany?

Line 173 - WTP presumably means willingness to pay but is never defined in the text

Line 187 - what is the difference between ICER and ICUR? This needs to be defined. Additionally, no primary outcome is ever stated, typically in cost effectiveness research, this is the ICER - is this what was considered the primary outcome for this manuscript?

Results

Line 226 - This paragraph should not be in the results section and instead should be moved to discussion.

Line 238 - again the distinction between ICER and ICUR needs to be made to make sense of what this data is conveying.

Discussion

Line 287 - In general I would reframe this pragraph to discuss how it is impractical to conduct randomized prospective research evaluating an outcome with such a small incidence as developing ovarian cancer. It is ethical to randomize patients to different treatment or prevention strategies when there is clinical equipoise as is the case in this situation, so I would remove that argument from this paragraph.

Line 323 "OS would be cost effective even if no maintenance therapy for ovarian cancer was applied" what does this mean? Was maintenance therapy factored into the model? This is not mentioned anywhere else. Are you referring to chemoprophylaxis with combined oral contraception? or maintenance medications for ovarian cancer like PARP inhibitors or avastin? Please clarify.

Line 339 - the emphasis here needs to be placed on the fact that this is a model and due to its inputs, it can only predict and therefore definitive conclusions cannot be made from this data. Because all of this data regarding risk reduction is based on knowledge from serous histologies, you really cannot say there are differences in risk reduction based on different histologies, because we have no data in the model to give you a sense of how this would affect someone with a clear cell carcinoma risk for example. I would remove this and make it clear this cannot be applied to a non serous setting, and furthermore, cannot be applied to patients without average background risk of ovarian cancer (ie. patients with BRCA mutations or at elevated risk of ovarian cancer due to family history).

Reviewer #2: Introduction. Key terminology missing here in this section is STIC (serous tubal intraepithelial carcinoma).

Ovarian cancer parameter (Table S1-6). One-size-fits-all type concept was used for ovarian cancer statistics / parameters. Based on histology types, stage distribution and prognosis are signifantly different. Also, salpingectomy is suggested to prevent certain epithelial type ovarian cancer. Ovarian cancer statistics for non-epithelial type (germ cell and sex cord-stromal) would not apply here and needed to be distinguished.

Cost effective threshold. This may be asssesed for various threshold cutpoints. Ovarian cancer treatment has evolved significantly in the recent years. More modern and effective treatments via biological targeted therapy is significantly expensive compared to systemic chemotherapy alone as shown in Table S10. These including anti-angiogenc therapy as well as synthetic lethalty-targeting poly [ADP]-ribose inhibitor matintenance therapy. In addition, universal genetic testing for individiual with epithelial ovarian cancer diagnosis as well as companion diagnostic testing became standard care in the management, and consideration of this testing cost also needed to be accounted in the threshold evaluation.

Cesarean delivery. This was only described in supplemental table. As salpingectomy is frequently performed at cesarean delivery, clarification of non-gyn surg in the main text would be useful otherwise readers have to look supplemental file to find the word cesarean.

Reviewer #3: Thank you for allowing me to review this relevant state transition analysis. The paper is well written and concise. Based on the results of this study, health policy maker will have to consider the implementation and re-imbursement of opportunistic salpingectomy (OS) as an add-on procedure in all abdominal surgical interventions. Congratulations.

The title 'Turning the tide on ovarian cancer' is though a bit too catchy in my view.

The figures need revision: Figure 1 remained incomprehensible for me even after reading the legend multiple times. In figure 2, the y-axis is labeled with risk per year but I am unsure if you mean risk for surgery or risk for ovarian cancer. Figure 3 is a tough one as well as I could not read the y-axis nor understand the abbreviation WTP.

Reviewer #4: Thank you to the authors for submitting this paper. I think the paper has the potential for publication in this journal, however there are several important changes I would make before publication. I have presented them in the order in which they appear in the paper.

Line 41: You report ICERs as your primary outcome measure. What relevance to they have in the German health care system? I'm aware that there is not an official threshold...

Line 89: Could you please further justify your model structure? There seems to be a large number of health states? Other semi markov models I've seen in the literature have used relatively simple model structures (progression free, post-progression, death). Does it need to be that complicated?

Line 120: What is the evidence on which the assumption that mortality returns back to background after 10 years based on?

Line 137-Line 140: The justification for these parameters seems a little suspect. Please further expand on the justification.

Line 150: A utility value of 1 should not be used for healthy states. The age-adjusted population average should be used. This is common practice in the literature.

Line 158: Please provide further evidence that the costs of the interventions should not be included.

Line 179: Please provide further justification that a one year cycle length is appropriate in this instance. What is this based on? Previous models in this area I've seen have used monthly cycles.

Line 179: Was half cycle correction used? If not, why not?

Line 188: Please define what you mean by "stable".

Line 192: Transition probabilities should be included in the sensitivity analysis, it doesn't matter if they are based on real-world population data or not.

Line 211: Please comment on why the prevalence may be lower in the model compared to the published data.

Table 1: Is is possible to use a consistent number of decimal places for the various parameters/results?

Line 273: A probabilistic sensitivity analysis needs to be included in analyses such as this in order to fully explore the uncertainty in the model.

Line 350: How important is the lack of discrimination between sub types of ovarian cancer? How would the inclusion of these sub-types impact the results?

I will comment further on the results in the next round of peer review once a PSA has been provided.

---

* We ask every co-author listed on the manuscript to fill in a contributing author statement, making sure to declare all competing interests. If any of the co-authors have not filled in the statement, we will remind them to do so when the paper is revised. If all statements are not completed in a timely fashion this could hold up the re-review process. If new competing interests are declared later in the revision process, this may also hold up the submission. Should there be a problem getting one of your co-authors to fill in a statement we will be in contact. Please do not add or remove authors without first discussing this with the handling editor. You can see our competing interests policy here: http://journals.plos.org/plosmedicine/s/competing-interests .

* Please upload any figures associated with your paper as individual TIF or EPS files with 300dpi resolution at resubmission; please read our figure guidelines for more information on our requirements: http://journals.plos.org/plosmedicine/s/figures . While revising your submission, please upload your figure files to the PACE digital diagnostic tool, https://pacev2.apexcovantage.com/ . PACE helps ensure that figures meet PLOS requirements. To use PACE, you must first register as a user. Then, login and navigate to the UPLOAD tab, where you will find detailed instructions on how to use the tool. If you encounter any issues or have any questions when using PACE, please email us at PLOSMedicine@plos.org.

* Please ensure that the paper adheres to the PLOS Data Availability Policy (see http://journals.plos.org/plosmedicine/s/data-availability ), which requires that all data underlying the study's findings be provided in a repository or as Supporting Information. For data residing with a third party, authors are required to provide instructions with contact information (web or email address) for obtaining the data. Please note that a study author cannot be the contact person for the data. PLOS journals do not allow statements supported by "data not shown" or "unpublished results." For such statements, authors must provide supporting data or cite public sources that include it.

* At this stage, we ask that you include a short, non-technical Author Summary of your research to make findings accessible to a wide audience that includes both scientists and non-scientists. The Author Summary should immediately follow the Abstract in your revised manuscript. This text is subject to editorial change and should be distinct from the scientific abstract. Ideally each sub-heading should contain 2-3 single sentence, concise bullet points containing the most salient points from your study. In the final bullet point of 'What Do These Findings Mean?', please include the main limitations of the study in non-technical language. Please see our author guidelines for more information: https://journals.plos.org/plosmedicine/s/revising-your-manuscript#loc-author-summary .

FIGURES AND TABLES

SUPPLEMENTARY MATERIAL

REFERENCES

* Please ensure that journal name abbreviations match those found in the National Center for Biotechnology Information (NCBI) databases (http://www.ncbi.nlm.nih.gov/nlmcatalog/journals ), and are appropriately formatted and capitalised.

* Where website addresses are cited, please include the complete URL and specify the date of access (e.g. [accessed: 12/06/2024]).

STUDY TYPE-SPECIFIC REQUESTS

The following list is derived from Geoffrey P Garnett, Simon Cousens, Timothy B Hallett, Richard Steketee, Neff Walker. Mathematical models in the evaluation of health programmes. (2011) Lancet DOI:10.1016/S0140-6736(10)61505-X: 

* If pertinent, please provide a diagram that shows the model structure, including how the natural history of the disease is represented, the process and determinants of disease acquisition, and how the putative intervention could affect the system.

* Please provide a complete list of model parameters, including clear and precise descriptions of the meaning of each parameter, together with the values or ranges for each, with justification or the primary source cited and important caveats about the use of these values noted.

* Please provide a clear statement about how the model was fitted to the data, including goodness-of-fit measure, the numerical algorithm used, which parameter varied, constraints imposed on parameter values, and starting conditions.

* For uncertainty analyses, please state the sources of uncertainties quantified and not quantified [can include parameter, data, and model structure].

* Please provide sensitivity analyses to identify which parameter values are most important in the model. Uncertainty estimates seek to derive a range of credible results on the basis of an exploration of the range of reasonable parameter values. The choice of method should be presented and justified.

* Please discuss the scientific rationale for the choice of model structure and identify points where this choice could influence conclusions drawn. Please also describe the strength of the scientific basis underlying the key model assumptions.

* For studies that develop a prediction model or evaluate its performance, please ensure that the study is reported according to the TRIPOD statement (https://www.equator-network.org/reporting-guidelines/tripod-statement ) and include the completed checklist as Supporting Information. Please add the following statement, or similar, to the Methods: "This study is reported as per the Transparent Reporting of a Multivariable Prediction Model for Individual Prognosis Or Diagnosis (TRIPOD) statement (S1 Checklist)." For studies using machine learning, please use the TRIPOD-AI checklist. When completing the checklist, please use section and paragraph numbers, rather than page numbers.

* For the cost-effectiveness section of your manuscript, please consider the items outlined in the CHEERS guideline/checklist (available at https://www.equator-network.org/reporting-guidelines/cheers ).

---

## [Decision Letter · Decision Letter 2]

5 Dec 2024

Dear Dr. Kather,

Thank you very much for re-submitting your manuscript "Ovarian Cancer Prevention with Cost-Effective Opportunistic Salpingectomy—A Decision Analytic Assessment Across Abdominal Surgeries" (PMEDICINE-D-24-02003R2) for review by PLOS Medicine.

Thank you for your detailed response to the editors' and reviewers' comments. I have discussed the paper with my colleagues and the academic editor, and it has also been seen again by two of the original reviewers. The changes made to the paper were mostly satisfactory to the reviewers. As such, we intend to accept the paper for publication, pending your attention to the editors' comments below in a further revision. When submitting your revised paper, please once again include a detailed point-by-point response to the editorial comments.

[LINK]

In revising the manuscript for further consideration here, please ensure you address the specific points made by each reviewer and the editors. In your rebuttal letter you should indicate your response to the reviewers' and editors' comments and the changes you have made in the manuscript. Please submit a clean version of the paper as the main article file. A version with changes marked must also be uploaded as a marked up manuscript file. Please also check the guidelines for revised papers at http://journals.plos.org/plosmedicine/s/revising-your-manuscript for any that apply to your paper.

We ask that you submit your revision within 1 week (Dec 12 2024). However, if this deadline is not feasible, please contact me by email, and we can discuss a suitable alternative.

Please do not hesitate to contact me directly with any questions (atosun@plos.org). If you reply directly to this message, please be sure to 'Reply All' so your message comes directly to my inbox.

We look forward to receiving the revised manuscript.

Sincerely,

Alexandra Tosun, PhD

Associate Editor 

PLOS Medicine

plosmedicine.org

Comments from Reviewers:

Reviewer #1: Thank you for allowing me the opportunity to re-review this article. My comments have been addressed and I have no further feedback for the manuscript.

Reviewer #4: Thank you to the authors for thoroughly responding to my comments - I am happy to approve this paper for publication.

[LINK]

Requests from Editors:

DATA AVAILABILITY

Please note that the Data Availability statement in the online submission form and the manuscript (lines 10-11) do not match. For each data source used in your study:

TITLE

Please revise your title according to PLOS Medicine's style. Your title must be nondeclarative and not a question. It should begin with main concept if possible. "Effect of" should be used only if causality can be inferred, i.e., for an RCT. Please place the study design ("A randomized controlled trial," "A retrospective study," "A modelling study," etc.) in the subtitle (ie, after a colon).

ABSTRACT

The abstract section could benefit from additional detail. What does ‘all relevant health states’ mean? What were the actual case numbers used? What were the outcomes included? We also think that ‘OS at all suitable surgery’ is not 100% clear. You might consider changing it to ‘OS performed at any suitable surgerical intervention’ (or similar) or add more detail before listing the four different strategies. More general points to consider are listed below:

*Please ensure that all numbers presented in the abstract are present and identical to numbers presented in the main manuscript text.

*Please quantify the main results (with 95% CIs and p values).

*Please include the study design, population and setting, number of participants, years during which the study took place, length of follow up, and main outcome measures.

*Please include the important dependent variables that are adjusted for in the analyses.

*Please describe the main outcome measures.

1) l.33: Please change ‘evaluates’ to ‘evaluated’.

2) l.41, “Volume of eligible interventions in strategy I was 3.5 times greater than in strategy III.” – please present numbers here, not just ratios.

3) l.47: “wide range of input-parameters” – could you please add more detail?

4) l.50/51: Please only use words, such as "significant" (or “substantially”) if statistical significance was tested.

5) In the last sentence of the Abstract Methods and Findings section, please describe the main limitation(s) of the study's methodology.

AUTHOR SUMMARY

Please note that the main limitations of the study should be included in the final bullet point of ‘What Do These Findings Mean?’.

INTRODUCTION

Please address past research and explain the need for and potential importance of your study. Indicate whether your study is novel and how you determined that. If there has been a systematic review of the evidence related to your study (or you have conducted one), please refer to and reference that review and indicate whether it supports the need for your study.

1) ll.103-104, please change to: “representing 22% of cancer cases”

2) l.106: Please refer to high income countries rather than "developed" or "Western" countries.

3) ll.116-117: Please provide references.

4) l.120: Based on the references provided, the description “all over the world” seems rather exaggerated. Please revise and specfiy.

5) l.130: “Furthermore, no concept for financial coverage of the 0 additional time and cost for OS has been developed for statutory health insurance systems.” – is this universally true?

METHODS AND RESULTS

1) ll.145-146, we suggest changing to: “It is not currently recommended that women with an average risk of developing ovarian cancer undergo surgery solely as a means of preventing the disease through salpingectomy.”

2) l.160: When presenting age, please provide a unit, such as years (“ages of 35 and 45 years”). Please revise throughout.

3) l.172ff: Please replace “all” with “any”. Please revise throughout.

4) l.172: Please note that you alternate between American and British English (e.g. gynecologic and gynaecologic). Please choose one and revise accordingly.

5) l.180: “current practice in many countries” – would it be possible to specify by providing a number?

6) ll.209-211: We suggest including the description of the equation into the main text and removing the equation description.

7) l.247: Please define ‘PARP’ at first use.

8) l.267: Please define ‘QALY’ at first use.

9) l.270: Please specify what the n-number refers to.

10) ll.317-319, please change to: “With regard to ovarian cancer case numbers in Germany (n = 7180 in 2020)[52], this would correspond to 395 (strategy III), 703 (strategy II) and 1099 (strategy I) prevented cases per year.”

11) l.319: “Lifetime risk could be lowered to 1:102.” – in which scenario?

12) ll.321-322: And what about strategy I?

13) ll.323-324, please change to: “Without discounting, 0.00542 (III), 0.00956 (II) and 0.01369 (I) LYs and 0.00466 (III), 0.00822 (II) and 0.01178 (I) QALYs could be gained per person compared to strategy IV”. Please revise accordingly throughout (i.e. replace 'resp.' with 'and').

14) Table 1:

a) When presenting age, please provide a unit, such as years.

b) For, ‘Mean age at diagnosis’ and ‘Average age at OS’, please provide a range, e.g. SD.

c) Please briefly explain the four strategies below the table.

15) ll.357-358: Please spell out ‘comp’.

16) l.358: Please remove the word ‘respectively’.

17) l.369-371, please change to: “However, additional expenditures (positive ICER) would occur if the cost of OS exceeded €332.01 (strategy I), €330.23 (II), and €267.04 (III) (Figure 3D), corresponding to more than 20 min, 19.9 min, and 16.1 min of additional operating time, respectively.”

18) l.372: Please change ‘are’ to ‘were’.

19) l.373: Please change ‘resp.’ to ‘and’ as well as ‘is’ to ‘was’.

20) Figure 1: We feel that the figure would be more accessible if you avoided using abbreviations. If you decide to keep it as it is, please change 'HEBSO' to 'HE+BSO' in the figure description.

21) Figure 2: Please remove ‘respectively’ (line 536).

22) Figure 3:

a) Please write the title of the x-axis below the axis.

b) Please define 'QALY'.

c) Does 'base case' refer to Stage IV? Please clarify.

DISCUSSION

Please present and organize the Discussion as follows: a short, clear summary of the article's findings; what the study adds to existing research and where and why the results may differ from previous research; strengths and limitations of the study; implications and next steps for research, clinical practice, and/or public policy; one-paragraph conclusion.

1) Please remove any subheadings from the discussion section, including the conclusion subheading.

2) l.399/l.424: Please temper claims of primacy of results by stating, "to our knowledge" or something similar.

3) l.436: Please replace ‘higher’ with ‘advanced’.

4) l.444: Please remove ‘as its main result’.

REFERENCES

1) Where website addresses are cited, please use the word ‘accessed’ to specify the date of access (e.g. [accessed: 12/06/2024]).

SUPPLEMENTARY MATERIAL

In the published article, supporting information files are accessed only through a hyperlink attached to the captions. For this reason, you must list captions at the end of your manuscript file. You may include a caption within the supporting information file itself, as long as that caption is also provided in the manuscript file. Do not submit a separate caption file.

When SI files are contained with a single file:

Please label the file as ‘S1 Supporting Information’.

Please apply alphabetical labelling to each table and figure contained within the S1 file. For example, ‘Fig A’ to ‘Fig Z’ and ‘Table A’ to ‘Table Z’.

Plain text does not need to be labelled and can just be given a title as necessary. For example, ‘Statistical Analysis Plan’.

Please cite tables/figures as ‘Fig A in S1 Supporting Information’ and/or ‘Table A in S1 Supporting Information’, for example.

Please cite plain text as, ‘Statistical Analysis Plan in S1 Supporting Information’, for example.

When SI files are uploaded as separate files:

Please label tables as ‘S1 Table’ (so on) and figures as ‘S1 Fig’ (and so on).

Any additional documents (protocols/analysis plans etc.) can be labelled as ‘S1 Protocol’, for example. Please cite items as exactly as labelled.

SOCIAL MEDIA

To help us extend the reach of your research, please provide any social media handle(s) that would be appropriate to tag, including your own, your co-authors’, your institution, funder, or lab. Please enter in the submission form any handles you wish to be included when we post about this paper.

General Editorial Requests

2) Please ensure that the paper adheres to the PLOS Data Availability Policy (see http://journals.plos.org/plosmedicine/s/data-availability ), which requires that all data underlying the study's findings be provided in a repository or as Supporting Information. For data residing with a third party, authors are required to provide instructions with contact information for obtaining the data. PLOS journals do not allow statements supported by "data not shown" or "unpublished results." For such statements, authors must provide supporting data or cite public sources that include it.

---

## [Editor Report · Decision Letter 3]

18 Dec 2024

Dear Dr Kather, 

On behalf of my colleagues and the Academic Editor, James Derek Brenton, I am pleased to inform you that we have agreed to publish your manuscript "Ovarian cancer prevention through opportunistic salpingectomy during abdominal surgeries: A cost-effectiveness modelling study" (PMEDICINE-D-24-02003R3) in PLOS Medicine.

I appreciate your thorough responses to the reviewers' and editors' comments throughout the editorial process. We look forward to publishing your manuscript, and editorially there are only a few remaining minor stylistic points that should be addressed prior to publication. We will carefully check whether the changes have been made. If you have any questions or concerns regarding these final requests, please feel free to contact me at atosun@plos.org.

Please see below the minor points that we request you respond to:

1) Abstract, ll.37-39: Please move the sentence about the primary outcome to the Methods and Findings section (suggestion: line 48 after "...and IV) no implementation of OS.").

2) Abstract: On line 44, you say that the model is based on inpatient case numbers. However, as the main limitation on line 63, you wrote that the data came from the outpatient sector (we assume it should be 'inpatient' here). Please revise.

3) Abstract, ll.66-68, to reiterate the definition of strategy I, we suggest changing the last sentence to: “Based on a lifetime cost saving of €20.89 per capita if OS was performed at any suitable abdominal surgery, the estimated total healthcare cost savings in Germany could be more than €10 million annually.”

4) Discussion, l.506: ‘data from the outpatient sector’ – please check and revise according to comment #2 (and please check throughout the main text).

Before your manuscript can be formally accepted you will need to complete some formatting changes, which you will receive in a follow up email (including the editorial points above). Please be aware that it may take several days for you to receive this email; during this time no action is required by you. Once you have received these formatting requests, please note that your manuscript will not be scheduled for publication until you have made the required changes.

PRESS

We also ask that you take this opportunity to read our Embargo Policy regarding the discussion, promotion and media coverage of work that is yet to be published by PLOS. As your manuscript is not yet published, it is bound by the conditions of our Embargo Policy. Please be aware that this policy is in place both to ensure that any press coverage of your article is fully substantiated and to provide a direct link between such coverage and the published work. For full details of our Embargo Policy, please visit http://www.plos.org/about/media-inquiries/embargo-policy/ .

Sincerely, 

Alexandra Tosun, PhD 

Associate Editor 

PLOS Medicine